# Exercise Intervention Mitigates Pathological Liver Changes in NAFLD Zebrafish by Activating SIRT1/AMPK/NRF2 Signaling

**DOI:** 10.3390/ijms222010940

**Published:** 2021-10-10

**Authors:** Yunyi Zou, Zhanglin Chen, Chenchen Sun, Dong Yang, Zuoqiong Zhou, Xiyang Peng, Lan Zheng, Changfa Tang

**Affiliations:** Key Laboratory of Physical Fitness and Exercise Rehabilitation of Hunan Province, College of Physical Education, Hunan Normal University, Changsha 410012, China; 202020151346@hunnu.edu.cn (Y.Z.); zhanglinchen@hunnu.edu.cn (Z.C.); sunchenchen1022@hunnu.edu.cn (C.S.); yangdong@hunnu.edu.cn (D.Y.); zhouzuoqiong26@hunnu.edu.cn (Z.Z.)

**Keywords:** zebrafish, NAFLD, exercise, ROS, SIRT1/AMPK/NRF2

## Abstract

Non-alcoholic fatty liver disease (NAFLD) is a common disease that causes serious liver damage. Exercise is recognized as a non-pharmacological tool to improve the pathology of NAFLD. However, the antioxidative effects and mechanisms by which exercise ameliorates NAFLD remain unclear. The present study conducted exercise training on zebrafish during a 12-week high-fat feeding period to study the antioxidant effect of exercise on the liver. We found that swimming exercise decreased lipid accumulation and improved pathological changes in the liver of high-fat diet-fed zebrafish. Moreover, swimming alleviated NOX4-derived reactive oxygen species (ROS) overproduction and reduced methanedicarboxylic aldehyde (MDA) levels. We also examined the anti-apoptotic effects of swimming and found that it increased the expression of antiapoptotic factor *bcl2* and decreased the expression of genes associated with apoptosis (*caspase3*, *bax*). Mechanistically, swimming intervention activated SIRT1/AMPK signaling-mediated lipid metabolism and inflammation as well as enhanced AKT and NRF2 activation and upregulated downstream antioxidant genes. In summary, exercise attenuates pathological changes in the liver induced by high-fat diets. The underlying mechanisms might be related to NRF2 and mediated by SIRT1/AMPK signaling.

## 1. Introduction

Nonalcoholic fatty liver disease (NAFLD) is one of the most prevalent causes of chronic liver disease, and its morbidity continues to rise worldwide in the context of the obesity epidemic. NAFLD is a chronic liver disease with excessive fat accumulation in liver and accompanied by other complications, such as insulin resistance, type 2 diabetes, and cardiopathy [1]. As a continuum of liver abnormalities, NAFLD has an intricate pathway from nonalcoholic fatty liver (NAFL) to nonalcoholic steatohepatitis (NASH) [2,3]. NASH is a more aggressive and serious stage of inflammation and steatohepatitis, and is accompanied by fibrosis and cirrhosis [3]. The fibrosis stage of NASH is associated with an increased risk of mortality [4]. In general, a high caloric, high-fat diet, and sedentariness are responsible for the progression of NAFLD [5]. Excessive energy intake and lack of physical activity, leading to obesity, and excessive accumulation of lipids within hepatocytes are the dominant risk factors for NAFLD [5]. Currently, dietary intervention is thought to be the standard care for NAFLD and NASH, but it often fails to control the progression of the disease [1]. Moreover, there are no normative pharmacological treatments that could prevent the progression from NAFL to NASH, or reverse NASH once it has developed [6].

Reactive oxygen species (ROS) play a vital role in regulating cell growth and hepatocyte death in NAFLD [7]. Prolonged high-fat diets induce ROS accumulation in the liver and impair the normal physiological function of hepatocytes [8]. Antioxidant therapy for NAFLD, such as treatment with glutathione, could significantly decrease ALT levels in patients [9]. NADPH oxidase 4 (NOX4) is the main source of ROS generation [10]. Targeting NOX4- or NOX4-derived ROS could be a potential therapeutic strategy for NAFLD [11,12]. Nuclear factor erythroid-2-related factor 2 (NRF2) is an important endogenous transcription factor that can defend cells against oxidative stress [13]. During NAFLD, a reduction in NRF2 levels is observed with redox upregulation [14,15]. Pharmacological activation of NRF2 restores injured liver tissue and attenuates oxidative stress in NAFLD animal models [16,17,18]. Therefore, antioxidative interventions in patients with NAFLD could prevent oxidative stress by activating Nrf2 and its downstream genes.

Regular exercise is an effective non-pharmaceutical therapy for NAFLD, and the underlying mechanism is related to the metabolic sensors SIRT1/AMPK [19]. A previous study revealed that exercise could stimulate SIRT1/AMPK signaling, which can both improve the pathogenesis and prevent the development of NAFLD [20]. The activation of SIRT1/AMPK signaling induced by exercise could decrease lipid accumulation, increase energy expenditure, limit de novo lipogenesis (DNL), and upregulate fatty acid metabolism [21,22]. SIRT1/AMPK signaling also attenuates inflammation [23], and may be a potential therapeutic target for reducing antioxidative stress and ameliorating oxidative impairment. SIRT1/AMPK signaling regulates NRF2 in pneumonia pathogenesis, cardiovascular diseases, and diabetic nephropathy [24,25,26]. Therefore, we speculate that exercise intervenes in NAFLD by exerting antioxidative effects, and that the molecular mechanism is related to SIRT1/AMPK/NRF2 signaling.

Zebrafish are a classical model system used to study liver initiation and development during embryogenesis [27]. Because of the similar hepatic cellular composition, function, signaling, response to injury, and cellular processes that regulate liver disease between humans and zebrafish, zebrafish have been used to explore liver diseases, liver cancer, and regeneration [28]. Due to their vigorous swimming capability, researchers have established swimming tunnel systems in zebrafish models for studying exercise physiology [29]. Previous studies indicate that zebrafish are a suitable model to investigate the mechanisms by which exercise could ameliorate NAFLD [30,31,32]. Therefore, in the present study, we established a zebrafish high-fat diet (HFD) NAFLD model and exercised the HFD zebrafish to explore the role of SIRT1/AMPK/NRF2 signaling in the observed beneficial effects of exercise. These findings could form the basis of a novel complementary therapeutic strategy for NAFLD, which could prevent the development of more serious disease pathology.

## 2. Results

### 2.1. Swimming Exercise Reduces Body Weight Gain and Lipid Accumulation in HFD Zebrafish

After 12 weeks, both female and male HFD zebrafish had significantly increased body weight compared to normal diet (ND) zebrafish. However, high-fat diet plus exercise (HEX) inhibited excessive body weight gain (Figure 1A,B). Oil Red O staining revealed that HFD zebrafish livers had more lipid droplets than ND zebrafish livers, and that the HEX group had fewer lipid drops than the HFD group (Figure 1C,D). The HEX group also had improved fasting blood glucose, liver triglyceride, and cholesterol levels compared to the HFD group (Figure 1E–G). These results suggest that exercise can mitigate lipid homeostasis disorders caused by excessive fat intake.

### 2.2. Swimming Exercise Activates SIRT1/AMPK Signaling and Improves Lipid Metabolism Disorders in HFD Zebrafish Livers

The SIRT1/AMPK signaling pathway is closely related to lipid metabolism, and previous research has suggested that it may contribute to the pathogenesis of NAFLD [33,34]. First, we showed that HFD zebrafish had strong inhibition of SIRT1 expression and AMPK phosphorylation compared to ND zebrafish, indicating that a high-fat diet significantly suppressed SIRT1/AMPK signaling. Conversely, swimming exercise upregulated SIRT1 expression and AMPK phosphorylation in HEX zebrafish (Figure 2A). Excessive body weight gain and fat accumulation induced by an HFD can be attributed to lipid metabolism disorders. We therefore next measured the mRNA expression of lipid metabolism genes regulated by SIRT1/AMPK signaling. HEX zebrafish had comparatively lower levels of lipogenesis genes (*acaca*, *fasn*, *srebf1*, *pparg*) (Figure 2B) and higher level of fatty acid β-oxidation genes (*pgc1α*, *pparab*, *acox1*, *cpt1a*) (Figure 2C) than HFD zebrafish.

### 2.3. Swimming Exercise Ameliorates Liver Pathological Changes in HFD Zebrafish

Zebrafish livers were stained with HE to investigate pathological changes associated with NAFLD. After 12 weeks of HFD, the zebrafish livers developed macrovesicular steatosis, lobular inflammation, and hepatocellular ballooning, and had significantly increased NAS scores, indicating the development of NAFLD (Figure 3A,B).

Moreover, proinflammatory cytokines (*il1β* and *tnfα*) were upregulated significantly in HFD group (Figure 4A). Swimming treatment attenuated these pathological changes in liver tissue, reducing the incidence of steatosis and cell ballooning and suppressing *il1β* and *tnfα* expression. The collagen content of the zebrafish livers was quantified, along with the protein levels of α-smooth muscle actin (α-SMA), to indicate the levels of fibrosis. These fibrotic markers were significantly increased in the livers of the HFD group. Swimming exercise prevented the HFD-induced fibrotic response in the HEX zebrafish livers, which had a lower collagen content and α-SMA protein expression compared to the HFD group (Figure 4B–D). These data suggest that HFD zebrafish could develop the pathology of NAFLD, and that swimming exercise could attenuate the pathogenic effects caused by the HFD.

### 2.4. Swimming Exercise Attenuates Hepatocyte Apoptosis in HFD Zebrafish

TUNEL staining was used to detect hepatocyte apoptosis. TUNEL-positive hepatic cells increased significantly in the HFD group compared to in the other two groups, and swimming exercise decreased the number of apoptotic cells compared to the HFD group (Figure 5A,B). Moreover, zebrafish in the HFD group had upregulated genes related to apoptosis (*caspase3*, *Bax*) and a downregulated anti-apoptosis gene (*bcl2*), while exercise appeared to reduce the rate of apoptosis (Figure 5C). Taken together, these data suggest that swimming protected hepatocytes from apoptosis.

### 2.5. Swimming Exercise Protects HFD Zebrafish Livers from Oxidative Stress

Oxidative stress affects many physiological processes under pathological conditions, including cell survival [35]. The progression of simple steatosis to NASH can be prevented by decreasing ROS production [36]. As NOX4 is the main source of ROS, we examined the protein levels of NOX4 in the HFD and HEX groups. NOX4 was significantly upregulated in the HFD and HEX groups compared with the ND group, while the expression of NOX4 was slightly decreased in the HEX group compared to in the HFD group (Figure 6A). We next investigated the antioxidative effects of swimming exercise in HFD zebrafish. Dihydroethidium (DHE) staining revealed that swimming exercise significantly decreased the high hepatic ROS levels associated with a HFD (Figure 6B,C). In addition, the elevated malondialdehyde (MDA) level induced by the HFD were also decreased in fish in the HEX group (Figure 6D). The above results indicate that exercise can alleviate oxidative stress in the livers of zebrafish.

### 2.6. Swimming Exercise Upregulates the Expression and Function of NRF2 in HFD Zebrafish

Accumulating evidence indicates that enhanced APMK activity is related to increased NRF2 expression and its antioxidative function. AMPK phosphorylation can directly promote NRF2 nuclear accumulation [37] or upregulate AKT phosphorylation and activate NRF2 [38]. In the present study, significantly lower expression levels of p-AKT and NRF2 were detected in the livers of HFD zebrafish compared to in ND and HEX zebrafish livers. Swimming attenuated the degradation of p-AKT and NRF2 induced by HFD (Figure 7A). Consistent with the observed downregulation of NRF2 expression, NRF2 downstream antioxidant genes *ho-1*, *nqo1*, and *cat* expression were significantly decreased in HFD zebrafish livers compared to in ND and HEX zebrafish livers (Figure 7B). These results suggest that exercise attenuates oxidative stress that is associated with the activation of NRF2 signaling in HFD zebrafish.

## 3. Discussion

In the present study, we adopted a zebrafish model to study the beneficial effects of swimming exercise on the pathological changes induced by high-fat diets. Zebrafish are an excellent exercise model for investigating NAFLD-related regulatory mechanisms. Zebrafish in the HFD group accumulated liver features that resemble the pathology of NAFLD, including lipid accumulation, hepatosteatosis, inflammation, fibrosis, apoptosis, and high levels of ROS. Swimming exercise mitigated these histopathological changes in the livers of the HEX zebrafish. This outcome is associated with the activation of SIRT1/AMPK signaling-mediated NRF2 upregulation in the liver.

SIRT1 plays a beneficial role in modulating hepatic lipid metabolism, and the activation of SIRT1 hinders the progression of fatty liver diseases [39]. AMP-activated protein kinase (AMPK) and SIRT1 are evolutionarily conserved partners with similar functions [40]. SIRT1/AMPK signaling is an energy-sensitive pathway that regulates various physiological reactions, including catabolic metabolism, angiogenesis, cell survival, and insulin resistance [40,41], and can antagonize the pathogenesis of NAFLD [42,43,44]. Both in vivo and in vitro NAFLD models have reported suppressed SIRT1/AMPK in NAFLD [33,45,46]. Activation of SIRT1/AMPK can improve liver injury in NAFLD by regulating lipid metabolism [47,48], relieving inflammation [49], and attenuating endoplasmic reticulum stress and insulin resistance [50]. Moreover, the Sirt1/AMPK pathway was identified as a regulator of exercise-induced adaptation in the liver [20,22,51]. We measured the protein levels of Sirt1 and the p-AMPK/AMPK ratio in zebrafish livers and found that HFD zebrafish had reduced Sirt1 expression and a decreased p-AMPK/AMPK ratio. In the HEX group, swimming exercise attenuated the suppression of the SIRT1/AMPK pathway induced by high-fat diets. These results indicate that swimming intervention could activate SIRT1/AMPK signaling and reduce liver damage caused by a high-fat diet.

Many studies have attempted to clarify the pathogenesis of NAFLD, and most of these have confirmed that simple symptomatic steatosis in the liver is the initial stage of NAFLD [52]. Unhealthy dietary habits, such as high fat, high carbohydrate, and high fructose diets, are thought to be the predominant cause of hepatic fat accumulation [53,54]. In the present study, the HFD group had significantly more lipid droplets, and had high hepatic TG and TC levels. The HEX group had reduced lipid accumulation compared to the HFD group. Further investigation of the downstream genes of the SIRT1/AMPK pathway that are related to lipogenesis and fatty acid β-oxidation revealed that excessive lipid intake increased the expression of lipogenesis genes (*acaca*, *fasn*, *srebf1*, and *pparg*) and reduced the expression of fatty acid β-oxidation genes (*pgc1α*, *pparab*, *acox1*, and *cpt1a*) in the livers of the HFD group. Conversely, swimming exercise significantly increased the expression of lipogenesis genes and promoted the expression of β-oxidation genes in the livers of the HEX group. These results confirm that the reduction of lipid accumulation and the maintenance of lipid homeostasis that are induced by swimming exercise contribute to the positive response to exercise in the zebrafish liver.

In previous studies, NAFLD patients were divided into NAFL and NASH groups according to their histological traits [55,56,57]. NASH is identified as a destructive stage of NAFLD, which is associated with hepatic steatosis, inflammation, ballooning, degeneration of hepatocytes, and fibrosis [55]. To further confirm the protective effects of exercise on zebrafish livers, in the present study we showed that high-fat diets increased hepatic steatosis, inflammatory infiltration, and the number of balloon cells. These histological features were measured using a grading and staging system [57]. The zebrafish livers in the HFD group had higher NAS scores, and swimming exercise reduced the NAS scores in the HEX group. We compared the mRNA expression of *il1β* and *tnfα* in the three groups and showed that HFD zebrafish livers had significantly higher *il1β* and *tnfα* expression. *Il1β* is a pivotal proinflammatory cytokine that causes liver inflammation, steatosis, and injury [58,59,60]. *Tnfα* is another crucial proinflammatory factor that is associated with the pathogenesis of human NASH [61,62] and animal NAFLD models [62,63,64]. Activation of SIRT1/AMPK suppresses inflammation and releases *il1β* [45,65]. In the present study, consistent with the activation of SIRT1/AMPK, swimming alleviated inflammation by suppressing *il1β* and *tnfα* expression induced by a high-fat diet. HFD zebrafish livers had remarkable collagen deposition, with higher fibrosis scores and higher levels of α-SMA protein. The HEX group had lower fibrosis scores, with less expression of fibrotic markers compared to the HFD group. These results demonstrate that NAFLD zebrafish had similar pathological changes induced by high-fat diets to NAFLD patients and other animal models. Swimming exercise could exert beneficial effects on diseased zebrafish livers.

The excessive generation of ROS is an undesirable pathological event during NAFLD development and is the hallmark between NAFL and NASH [66]. ROS can induce the proliferation and migration of HSCs, collagen accumulation, and the formation of liver fibrosis [67,68]. NADPH oxidase 4 (NOX4)-derived ROS are the main source of oxidative stress in NAFLD, and they contribute to hepatic injury [69]. In the present study, a high-fat diet upregulated the protein expression of NOX4, with high levels of superoxide anion production and MDA observed in the zebrafish liver. Overproduction of ROS is a harmful process that causes injury to cellular components, including DNA, proteins, lipids, mitochondria, and membrane structures. ROS may contribute to hepatocyte apoptosis in the liver [70]. The release of ROS appears to be a critical event in oxidant-induced apoptosis [71]. Severe hepatic cell apoptosis occurred in the livers of the HFD group, with associated upregulation of apoptosis-related markers and cleaved-caspase3 protein levels. The mRNA expression of apoptotic factors (*caspase3*, *Bax*) increased and antiapoptosis factors (*bcl2*) expression decreased in NAFLD model zebrafish livers. Swimming exercise attenuated increased apoptosis-related marker expression and reduced hepatocyte apoptosis in the HEX group zebrafish livers. Swimming exercise decreased oxidative stress and apoptosis induced by ROS in zebrafish livers.

NRF2 is an antioxidant factor that exerts prominent protective and ameliorative effects on oxidative-associated liver diseases [72]. Activation of NRF2 exhibits positive feedback in impaired mouse livers induced by lipotoxicity [73,74,75]. NRF2 is a downstream target of SIRT1/AMPK signaling and functions as an antioxidant in various diseases [24,25]. AMPK phosphorylation facilitates the activation of the master antioxidant regulator NRF2 [76] and reduces the levels of free radicals by increasing HO-1 expression (a downstream factor of NRF2) [77]. Mechanistically, AMPK activation causes nuclear accumulation of Nrf2, probably by inhibiting its nuclear export, which suggests that AMPK is a regulator of NRF2 [37,78]. In addition, GSK-3β has been identified as an inhibitor of NRF2, either by retaining it in the cytoplasm or by increasing Nrf2 export [79]. AMPK activation could elevate AKT phosphorylation, subsequently inhibiting GSK-3β activity through Ser9 phosphorylation, and finally upregulating NRF2 and its downstream genes [80]. In the present study, we demonstrated that swimming exercise activated p-AKT and NRF2 expression, and upregulated Nrf2 downstream antioxidants *ho-1*, *nqo1*, and *cat* in the HEX group. This implies that swimming exercise increased NRF2 protein and antioxidant gene levels, resulting in less superoxide anion production and lower MDA levels than the HFD group. Taken together, our results show that swimming exercise could improve oxidative injury induced by high-fat diets in zebrafish livers by activating NRF2 mediated by SIRT1/AMPK signaling.

In summary, we applied a HFD zebrafish model of NAFLD to investigate the antioxidative effects of exercise on NAFLD. Zebrafish had similar histopathology compared to human and rodent models. Swimming exercise ameliorated the pathogenesis of NAFLD in the livers of model zebrafish. Exercise suppressed excessive ROS production in NAFLD model zebrafish via upregulation of NRF2, mediated by SIRT1/AMPK signaling. In view of the fact that exercise mimetics, such as AIACR, metformin, and resveratrol, also have the effect of activating SIRT1/AMPK signaling [81], it is worthy of further study to investigate and compare the alternative or combined effect of these drugs with exercise in the treatment of NAFLD. Our study provides new insights towards the design of personal training programs based on molecular indicators as a treatment for NAFLD.

## 4. Materials and Methods

### 4.1. Animal Models

NAFLD zebrafish were induced by HFD feeding. AB strain zebrafish, aged 6 months, were raised under 14 h of light at 28 °C under standard husbandry conditions. After a week of adaptation, zebrafish were randomly divided into three groups (*n* = 30 zebrafish/group): normal diet (ND), high-fat diet (HFD), and high-fat diet plus exercise (HEX). Zebrafish in the ND group received a low-fat diet containing 6% fat (TP1FM21051, Trophic Animal Feed High-Tech Co., Ltd., Nantong, Jiangsu Province, China) for 12 weeks. Zebrafish in the HFD group were fed a high-fat diet with 24% fat (TP1FM21050, Trophic Animal Feed High-Tech Co., Ltd., Nantong, Jiangsu Province, China) for 12 weeks. Zebrafish in the HEX group received a high-fat diet similar to the HFD group, but were subjected to swimming exercise for 12 weeks. After the experimental period, all zebrafish were fasted overnight before sacrifice for the collection of blood and liver tissue. All aspects of this research were conducted in accordance with the Chinese guidelines for animal welfare and experimental protocols. Approval was obtained from the Animal Experiment Administration Committee of Hunan Normal University (Changsha, China) (approval number: 2018/046).

### 4.2. Exercise Protocol

Swimming exercise was performed as previously described [18,29,31,32]. Before commencement of the exercise experiment, the body lengths (BL) of AB strain zebrafish were measured and the critical swimming speed (Ucrit) was calculated using a Loligo^®^ System (#SW10600) [29]. The zebrafish in the HEX group were placed in a swimming tunnel that compelled them to swim at a 6 × BL/s swimming speed (16 cm/s, approximately 40% Ucrit) for the first month, which was increased to 8 BL/s (22 cm/s, ~55% Ucrit) for the next two months. Zebrafish were exercised for 5 days per week. Except for during exercise, zebrafish in the HEX group were kept under the same conditions as the ND and HFD groups. During periods of exercise, zebrafish in the HEX group were transferred to the swimming tunnel, acclimated for 30 min, and exercised for 4 h per day.

### 4.3. Blood Analysis

After 3 months, 9-month-old zebrafish blood was collected (ND, *n* = 13; HFD, *n* = 10; HEX, *n* = 11). Fasting serum glucose levels were measured using a blood glucometer purchased from Yuwell (Yuwell 580, Beijing, China).

### 4.4. Liver Triglyceride (TG), Total Cholesterol TC, and Malondialdehyde (MDA) Levels

Five isolated liver tissue samples from each group were mixed to prepare a 10% homogenate. The homogenate was centrifuged at 2500–3000 rpm for 10 min, and the supernatant was collected. The protein concentration was determined using a BCA assay, according to the manufacturer’s instructions (Nanjing Jiancheng, Nanjing). TG and TC levels were measured using commercial kits (Changchun Huili Biotech Co., Ltd., Changchun, Jilin Province, China). MDA levels were measured using a malondialdehyde (MDA) assay kit (TBA method; Nanjing Jiancheng, Nanjing).

### 4.5. Histological Analysis of Liver Sections

Fresh liver tissue samples (*n* = 3) were fixed in 4% paraformaldehyde solution for 24 h, embedded in paraffin, and sliced into 4 µm section for hematoxylin-eosin (H&E) staining and Masson staining. The NAFLD activity score (NAS) was calculated according to the guidelines provided by the Pathology Committee of the NASH Clinical Research Network [57] as follows: steatosis (<5% = 0, 5–33% = 1, 33–66% = 2, > 66% = 3), lobular inflammation (none = 0, < 2 foci = 1, 2–4 foci = 2, > 4 foci = 3), and hepatocellular ballooning (none = 0, few = 1, prominent = 2). All features were scored in a blinded manner based on six fields of view per sample. The individual scores for each field of view were summed to calculate the NAS for each animal. Histological assessments were performed by a pathologist who was blinded to treatment. Oil Red O staining was performed to determine the lipid content in the liver tissue. The specimen chuck was fixed on the slicer, and the tissue surface was rough cut prior to slicing at 8–10 μm. Specimens were scored according to the amount of red (Oil Red O) staining using ImageJ software (version 5.0).

### 4.6. Apoptosis Assay

A TUNEL assay (*n* = 3) was performed on paraffin-embedded tissue slices to assess apoptosis. The nuclei were stained with DAPI (blue) under ultraviolet excitation, while TUNEL was labeled with CY3 fluorescein (red).

### 4.7. Determination of Liver ROS Production

Fresh liver tissues (*n* = 3) were prepared for frozen slides, and ROS stain (DHE) solution (D7008, Sigma, dilution 1:500) was added to the marked area. Nuclei were stained with DAPI. The ROS-positive area was stained red with fluorescein. The positive area was scored using ImageJ software (version 5.0).

### 4.8. RNA Isolation and Real-Time Quantitative PCR

RNA extraction and real-time quantitative PCR were performed as previously described (one sample mixed with liver tissues from two fish, *n* = 6) [82]. The primer sequences performed are listed in Appendix A. Gene expression levels were calculated by normalizing to β-actin using the 2^−ΔΔCt^ method.

### 4.9. Western Blotting Analysis

Two livers from each group were combined into one sample and total protein was extracted for western blot analysis (one sample mixed with liver tissues from two fish, *n* = 8).

Western blotting was performed as described previously [83]. The antibodies used were as follows: rabbit anti-β-actin antibody (1:2000, Proteintech, Wuhan, China), rabbit anti-α-SMA antibody (1:1000, Proteintech, Wuhan, China), rabbit anti-SIRT1 (1:1500, Proteintech, Wuhan, China), rabbit anti-NOX4 antibody (1:1000, Proteintech, Wuhan, China), rabbit anti-NRF2 antibody (1:1500, Proteintech, Wuhan, China), rabbit anti-phospho-AKT (Ser473) antibody (1:2000, Cell Signaling), rabbit anti-phospho-AMPK (Thr172) antibody (1:1000, Cell Signaling), rabbit anti-AMPK antibody (1:1000, Proteintech, Wuhan, China). Protein expression was normalized to that of β-actin.

### 4.10. Statistical Analysis

All statistical analyses were performed using GraphPad Prism 9.0. Differences between groups were assessed using one-way analysis of variance (ANOVA) and a Tukey post hoc test. Differences were considered significant at *p* ≤ 0.05. Values are expressed as the mean ± standard error.

## Figures and Tables

**Figure 1 ijms-22-10940-f001:**
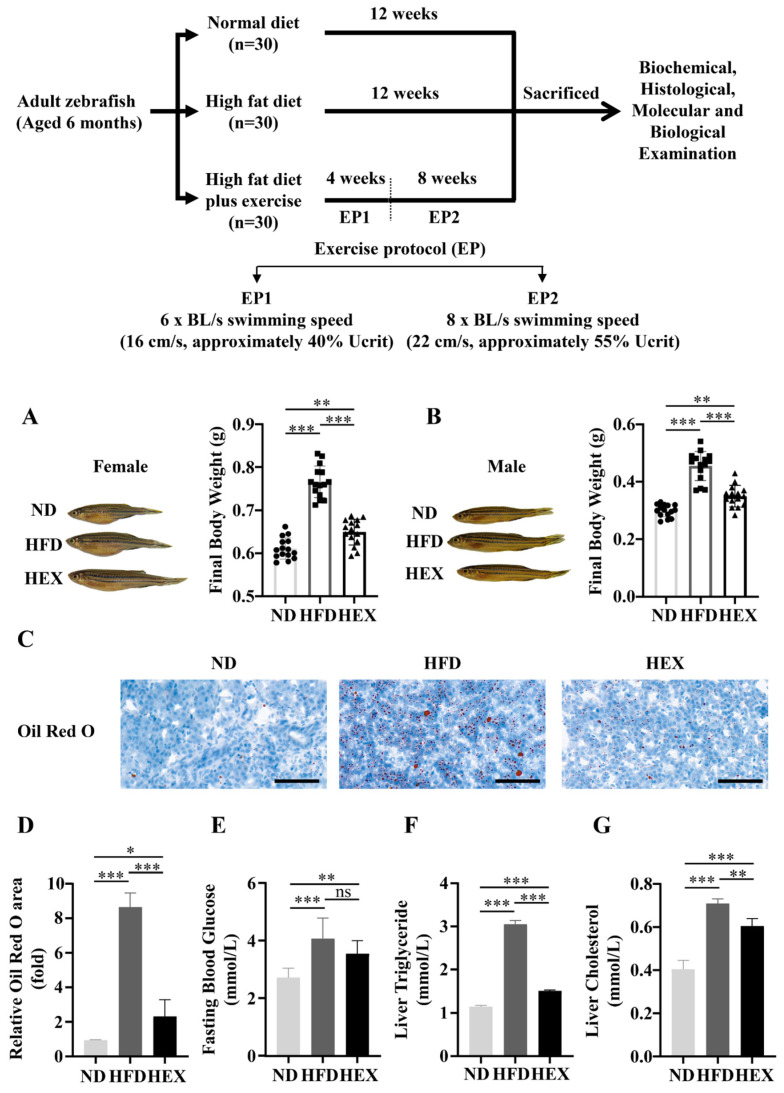
Zebrafish receiving a high exercise regimen (HEX) had reduced body weight gain and lipid accumulation compared to high-fat diet (HFD) zebrafish. Normal diet (ND) zebrafish represent the control group. (**A**,**B**) Morphology and body weight of zebrafish. (**C**) Oil Red O staining of zebrafish livers (*n* = 3). (**D**) Quantitation of Oil Red O staining. (**E**) Fasting blood glucose (ND, *n* = 13; HFD, *n* = 10; HEX, *n* = 11). (**F**) Liver triglyceride (*n* = 5). (**G**) Liver cholesterol (*n* = 5). The above experiments were carried out using 9-month-old zebrafish. *, *p* < 0.05, **, *p* < 0.01, ***, *p* < 0.001. Data represent the mean, and error bars represent SEM. Scale bar, 20 μm. NAFLD, non-alcoholic fatty liver disease; ND, normal diet; HFD, high fat diet; HEX, high-fat diet plus exercise.

**Figure 2 ijms-22-10940-f002:**
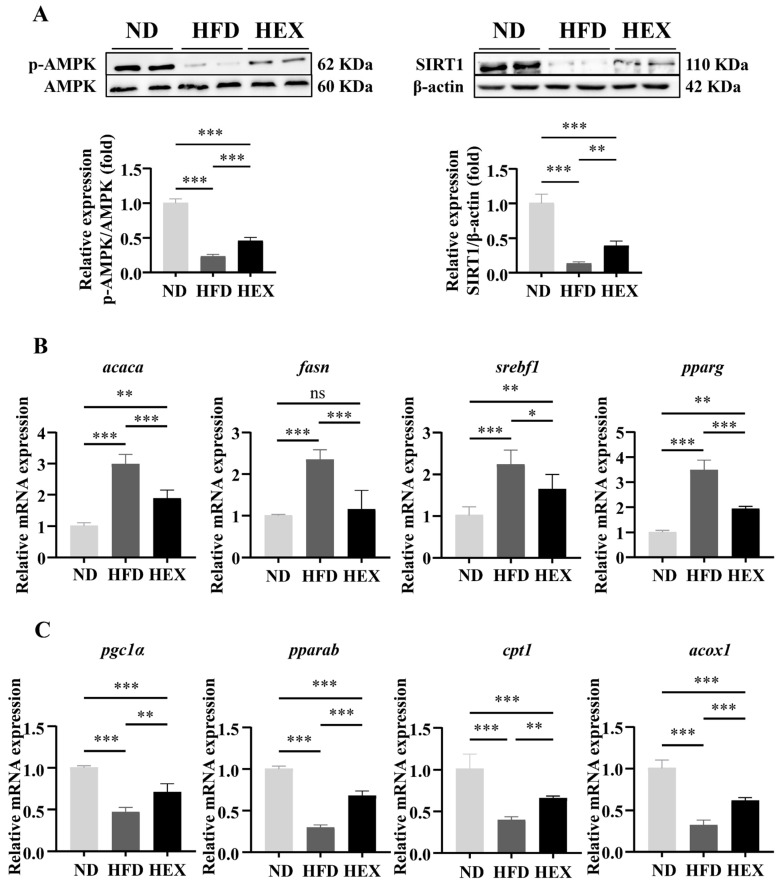
SIRT1/AMPK signaling and lipid metabolism in zebrafish livers. (**A**) Western blot representing SIRT1/AMPK signaling in 9-month-old zebrafish livers (*n* = 8). (**B**) Expression of lipogenesis-related genes (*n* = 6). (**C**) Expression of β-oxidation-related genes in 9-month-old zebrafish livers (*n* = 6). *, *p* < 0.05, **, *p* < 0.01, ***, *p* < 0.001. Data represents the mean, and error bars represent SEM. NAFLD, non-alcoholic fatty liver disease; ND, normal diet; HFD, high fat diet; HEX, high-fat diet plus exercise.

**Figure 3 ijms-22-10940-f003:**
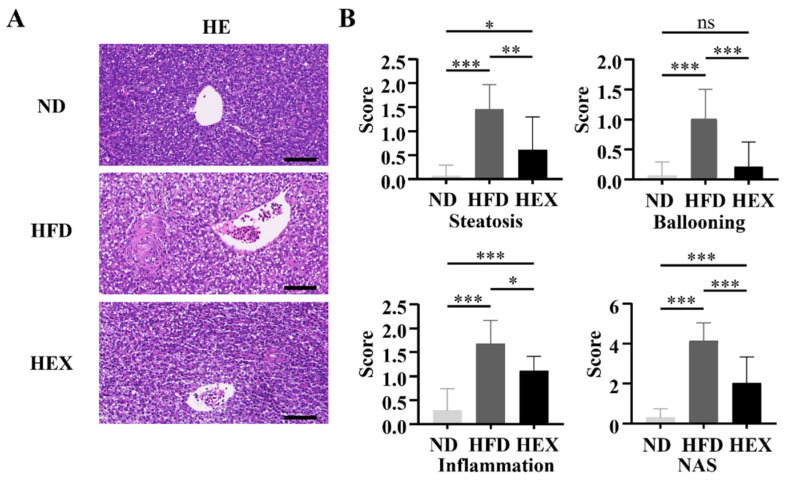
Quantitation of pathological changes in zebrafish livers. (**A**) H&E staining of 9-month-old zebrafish livers (*n* = 3). (**B**) NAFLD activity scores. *, *p* < 0.05, **, *p* < 0.01, ***, *p* < 0.001. Data represent the mean and error bars represent SEM. Scale bar, 20 μm. NAFLD, non-alcoholic fatty liver disease; ND, normal diet; HFD, high fat diet; HEX, high-fat diet plus exercise.

**Figure 4 ijms-22-10940-f004:**
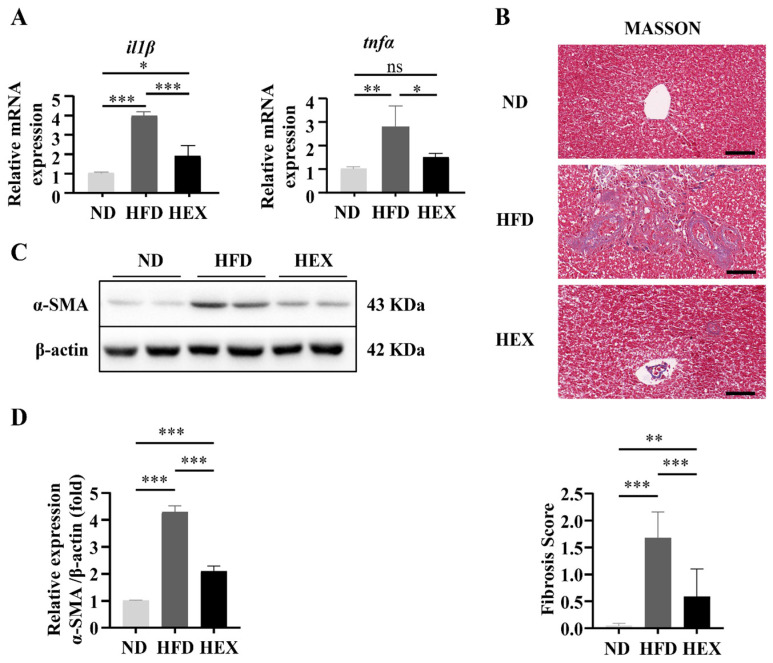
Quantitation of pathological changes in 9-month-old zebrafish livers. (**A**) Gene expression of *il1β* and *tnfα* (*n* = 6). (**B**) Masson’s staining and fibrosis score of zebrafish livers (*n* = 3). (**C**,**D**) The protein levels of α-smooth muscle actin (α-SMA) (*n* = 8). *, *p* < 0.05, **, *p* < 0.01, ***, *p* < 0.001. Data represent the mean and error bars represent SEM. Scale bar, 20 μm. NAFLD, non-alcoholic fatty liver disease; ND, normal diet; HFD, high fat diet; HEX, high-fat diet plus exercise.

**Figure 5 ijms-22-10940-f005:**
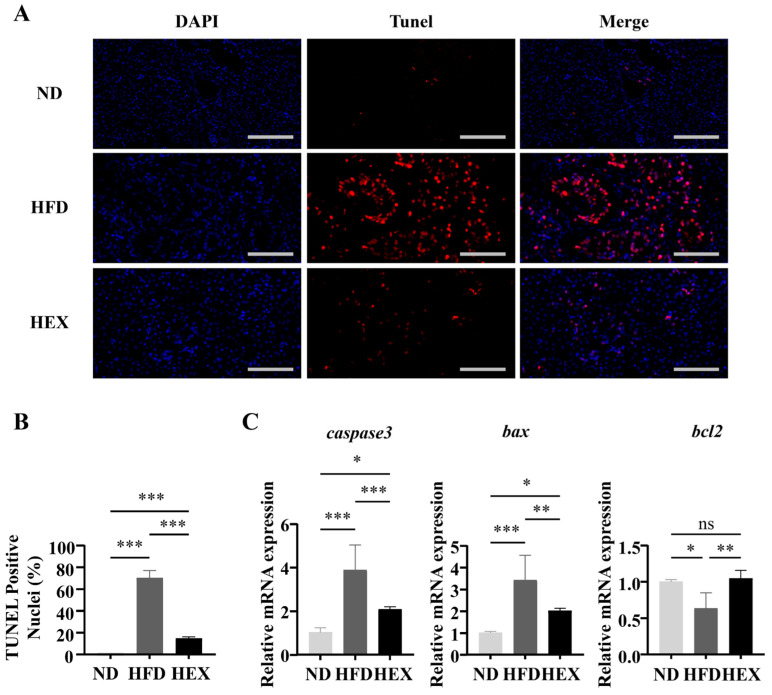
Hepatocyte apoptosis in 9-month-old zebrafish livers. (**A**) Tunel staining of zebrafish livers, and (**B**) Quantitation of Tunel-positive cells (*n* = 3). (**C**) Gene expression of *caspase3*, *bax*, and *bcl2* (*n* = 6). *, *p* < 0.05 **, *p* < 0.01, ***, *p* < 0.001. Data represent the mean, and error bars represent SEM. Scale bar, 20 μm. NAFLD, non-alcoholic fatty liver disease; ND, normal diet; HFD, high fat diet; HEX, high-fat diet plus exercise.

**Figure 6 ijms-22-10940-f006:**
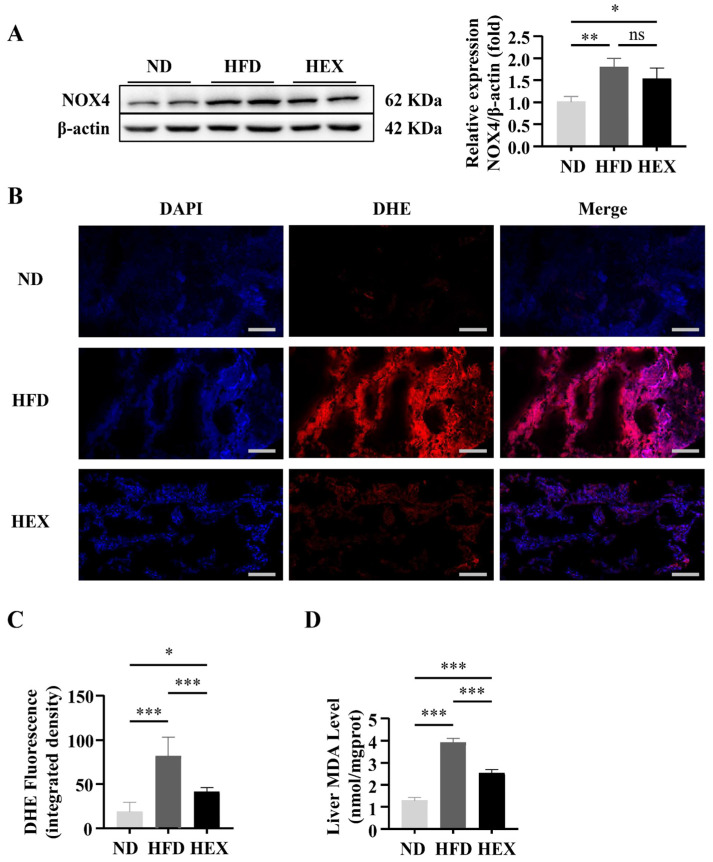
Oxidative stress in 9-month-old zebrafish livers. (**A**) NOX4 protein expression (*n* = 8). (**B**) Dihydroethidium (DHE) staining of zebrafish livers (*n* = 3), and (**C**) quantitation of DHE staining. (**D**) MDA levels in zebrafish livers (*n* = 5). *, *p* < 0.05, **, *p* < 0.01, ***, *p* < 0.001. Data represent the mean, and error bars represent SEM. Scale bar, 20 μm. NAFLD, non-alcoholic fatty liver disease; ND, normal diet; HFD, high fat diet; HEX, high-fat diet plus exercise.

**Figure 7 ijms-22-10940-f007:**
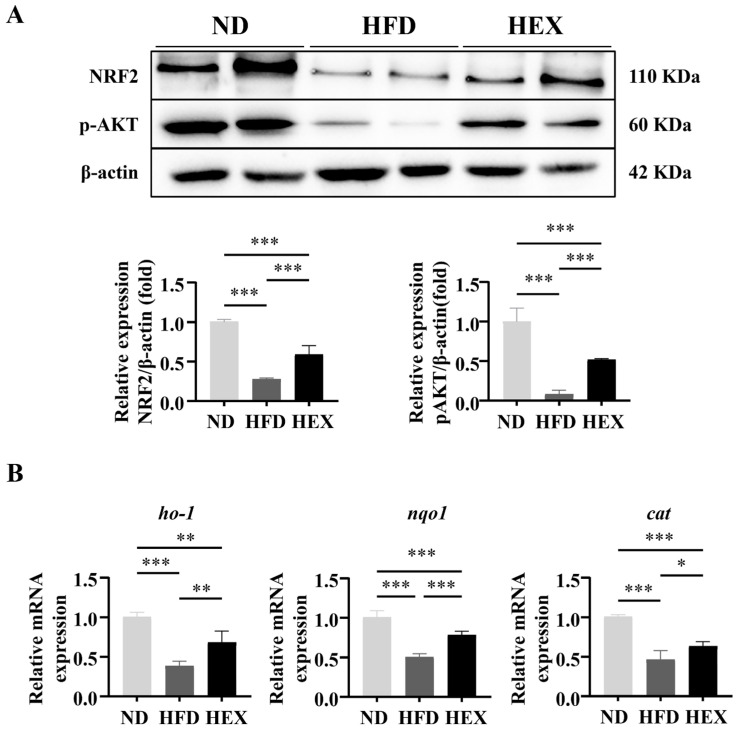
NRF2 signaling in 9-month-old zebrafish livers. (**A**) Western blot of p-AKT and NRF2 expression in zebrafish livers (*n* = 8). (**B**) Gene expression of *ho-1*, *nqo1*, and *cat* (*n* = 6). *, *p* < 0.05, **, *p* < 0.01, ***, *p* < 0.001. Data represent the mean, and error bars represent SEM. NAFLD, non-alcoholic fatty liver disease; ND, normal diet; HFD, high fat diet; HEX, high-fat diet plus exercise.

## Data Availability

The datasets generated and/or analyzed in the present study are available from the corresponding author upon reasonable request.

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
