# Peer review of "Exercise Intervention Mitigates Pathological Liver Changes in NAFLD Zebrafish by Activating SIRT1/AMPK/NRF2 Signaling"

_ijms, 2021, doi:10.3390/ijms222010940_

Round 1

Reviewer 1 Report

The authors here utilize zebrafish model to systematically evaluate the effects of exercise on the liver function especially in pathologies like NAFLD. The authors also highlight how a few nutrient sensing and signaling pathways such as SIRT1/AMPK/NRF2 play a crucial role in this process. This paper is of interest not only to the researchers in basic sciences, but also to clinicians who would want to determine proper therapeutic opportunities to combat and ameliorate NAFLD related pathologies. To make these studies more precise and convincing the authors need to address the following items:

Major comments:

  • In the introduction the authors should try to highlight more about the pathological conditions and clinical implications of these types of disorders. This review can be useful to highlight such studies PMID: 25575111.
  • As a first figure the authors should provide a summary of experimental procedures/design including the exercise regimen used which will help the readers navigate the paper easily.
  • Fig 2 A has no indication of which samples correspond to which bolts. Additionally, it will be good if the authors break this figure into 2 panels, one showing SIRT1 and another AMPK. Also, in figure 2 A the beta-actin levels don’t look uniform, please address that.
  • In the figure 2 panel also include a HDAC assay and an AMPK phosphorylation colorimetric assay to determine the enzymatic activity of each condition. This will further solidify the authors claims.
  • Since, the authors discuss how oxidative stress plays a crucial role in the entire process it would be good to include a panel where they determine the glutathione levels (oxidized vs reduced) for each condition. Since, glutathione plays a crucial role in oxidative stress and NAFLD (PMID: 28789631).
  • The authors mention the crucial role of SIRT1/AMPK/NRF2 related pathways in this entire process, it would be good to evaluate the effects of various drugs that modulate these pathways which could include either single or combinations of drugs like resveratrol, Hydralazine, Metformin, PPAR modulators and antioxidants and carry out head-to-head evaluations of these therapies when compared to exercise.

Minor comments:

  • For all the figures please include the stages/ age of zebrafish utilized in the representative figure.
  • The resolution of Fig 7 seems a bit off, please fix that.

Reviewer 2 Report

In this work the authors used an in vivo model of NAFLD, zebrafish fed with HFD, to investigate the molecular/cellular changes associated to exercise. They found that exercise activated SIRT1/AMPK signaling, regulating the expression of gene involved in lipogenesis and beta-oxidation;  enhanced  NRF2 pathway and upregulated downstream antioxidant  genes; decreased lipid accumulation and improved pathological changes in the liver of HFD animals. The results suggest that exercise, at least in zebrafish NAFLD model, counteracts the pathological events caused by HFD diet by a well-defined mechanism involving mainly SIRT1/AMPK and NRF2 pathways.  

Comments:

This work is well organized and written but I have some concerns regarding the methods and figures quality.

1) figure 5: DAPI signal is difficult to detect in control (ND). I’d also expect to find some apoptotic cell in physiological condition ( ND), while the representative image appear totally black.

2) same for  figure 6. Figures are not of good quality. DHE fluorescence of control is absent (I can see only the bacground) and DAPI staining is almost undetectable in all the figures showed. I’d expect to see a basal level of  ROS staining in control animals. I suggest to assemble a new panel with high quality figures .

3) figure 2A: Western blot image lacks sample names (ND, HFD and HEX)

4) please correct “AB staining”  in the methods (4.1)

5) It is not clear how many samples/replicates the authors have used for each experiment (Es. Western blot: authors show two replicates of each condition : they used only two biological replicates for quantifications? Please specify in methods and figures the information). Same for others experiments (qRT-PCR, DHE, tunel etc).

Round 2

Reviewer 1 Report

Thank you so much for updating the manuscript based on the suggestions. I fully understand the short time frame and the inability to perform some of these experiments. Although I still have a slight concern with the number of animals used for these studies which is around 3 to 4 for most of the experiments. It would be essential for the authors to justify in results or introduction the use of such low number of animals for the studies especially considering the model organism here is zebrafish, which you can have enough animals usually for all the studies. Also, hopefully for the future studies the authors can include some of the suggested experiments. 
